# Association between vitamin D deficiency and benign paroxysmal positional vertigo (BPPV) incidence and recurrence: a systematic review and meta-analysis

Heather Wood,[1] Karolina Kluk ![ORCID] ,[2] Ghada BinKhamis ![ORCID] [2,3]

¹College of Medical and Dental Sciences, University of Birmingham, Birmingham, UK
²Manchester Centre for Audiology and Deafness, The University of Manchester, Manchester, UK
³Cochlear Implant Center, King Fahad Medical City, Riyadh, Saudi Arabia

**Correspondence to**
Dr Ghada BinKhamis;
gbinkhamis@gmail.com

## ABSTRACT

**Objectives** The objective of this study is to determine the relationship between serum vitamin D level and the risk of developing benign paroxysmal positional vertigo (BPPV) incidence and recurrence in countries in the Northern Hemisphere.

**Design** Systematic review and meta-analysis.

**Data sources** PubMed, Scopus and Web of Science databases were searched for studies published between January 2000 and February 2023.

**Eligibility criteria for selecting studies** Participants located in the Northern Hemisphere aged 18 or over with at least one episode of BPPV, serum 25-hydroxyvitamin D levels measured and reported, no comorbidities or history of vitamin D supplementation.

**Data extraction and synthesis** Data extraction and synthesis were performed by a single reviewer and checked by a second reviewer. Inclusion and exclusion criteria and risk of bias were assessed by two independent reviewers using the Newcastle Ottawa Tool for Cohort studies and Risk of Bias Assessment Tool for Nonrandomised Studies checklist for case–control studies. Meta-analysis was conducted using random effects models. Standard mean difference with a 95% CI was used to measure the relationship between vitamin D level and BPPV.

**Results** The 35 articles identified by the literature search reported data of 9843 individuals. 19 studies (7387 individuals) were included in the BPPV incidence meta-analysis while 7 studies (622 individuals) were included in the BPPV recurrence meta-analysis. Lower serum vitamin D levels were found in BPPV incidence compared with controls, but the relationship between vitamin D levels in recurrent BPPV compared with non-recurrent disease remained uncertain.

**Conclusion** Results of this systematic review and meta-analysis demonstrated a negative correlation between serum vitamin D and BPPV incidence, while any relationship between serum vitamin D and BPPV recurrence remained uncertain. Risk of bias analysis revealed evidence of variable quality. There were insufficient data available to evaluate seasonal relationships between serum vitamin D and BPPV. Given the potential for this as a confounding factor, future research should aim to investigate this further.

## STRENGTHS AND LIMITATIONS OF THIS STUDY

⇒ This is a comprehensive systematic review of vitamin D and benign paroxysmal positional vertigo (BPPV) covering literature published between January 2000 and February 2023.
⇒ The meta-analyses include a large number of individuals (7510).
⇒ It was not possible to account for the impact of seasonality on both serum vitamin D levels and incidence and recurrence rates of BPPV as the majority of studies did not report the dates or months when patients were seen.
⇒ Data were not available relating to factors potentially related to both BPPV and vitamin D such as subject occupation, comorbidities and symptom severity.
⇒ High-quality research is currently limited, and a few of the identified studies in this systematic review were from journals with unclear peer-review processes or non-impact journals.

**PROSPERO registration number** CRD42021271840.

## INTRODUCTION

The impact of vitamin D on audiovestibular health has been a research area of interest in recent years, with low serum levels of vitamin D associated with otological pathologies including benign paroxysmal positional vertigo (BPPV).[1] Due to vitamin D's role in skeletal mineral homeostasis, it has been suggested that audiovestibular pathology may be related to high bone turnover rates in the temporal bone and/or at a systemic level.[2] Another hypothesis suggests that inner ear pathology including BPPV may be secondary to an inflammatory or autoimmune response following viral infection, and that the immunomodulatory functions of vitamin D may be protective against this secondary response.[3]

There are two major bioavailable forms of vitamin D; vitamin $D_2$ (also known as

ergocalciferol) and vitamin $D_3$ (cholecalciferol). Ergo-calciferol can be ingested via diet or supplementation, while vitamin $D_3$ is largely synthesised in human skin following exposure to ultraviolet B light (and may also be ingested via dietary supplements or animal-based foods).[4] The bioactive form of vitamin $D_3$ is known to play an important role in calcium homeostasis and bone health; increasing the absorption of dietary calcium and phosphorus which in turn affects the regulation of bone mineral density.[5] Vitamin $D_3$ is hydroxylated in the liver into 25-hydroxyvitamin D (calcifediol), before being processed further by the kidneys, converting it to its bioactive form (calcitriol).[5] Levels of the calcitriol precursor 25-hydroxyvitamin D (calcidiol) can be measured from blood samples and used to indicate the overall abundance of vitamin D within the body.

BPPV is a common vestibular disorder, characterised by positional vertigo and nystagmus elicited by head movements. It is caused by the displacement of small crystals of calcium carbonate known as otoconia into one of the semicircular canals (most commonly a posterior canal). Displacement of otoconia can occur following head trauma, however, most cases are idiopathic and the exact pathophysiology of BPPV is currently not known.[6]

The relationship between vitamin D and BPPV is currently a popular topic of research, and a number of systematic reviews have been published in recent years indicating that BPPV incidence is associated with vitamin D deficiency,[7 8] although this relationship was not supported by an earlier review.[9] There is mixed evidence regarding the relationship between vitamin D levels and BPPV recurrence, with some reviews finding lower levels of the vitamin in patients with recurrent BPPV,[9 10] while others have observed no significant difference in vitamin D levels and BPPV recurrence.[11] Supplementation in BPPV patients with low vitamin D levels has also been shown to result in prolonged improvement in symptoms and fewer episodes of recurrent BPPV.[12 13]

There is currently uncertainty relating to how seasonal patterns of vitamin D and BPPV incidence are related. In countries within the Northern Hemisphere, vitamin D levels fluctuate in a seasonal pattern—serum vitamin D levels are highest within the population of these countries during summer months and lowest during winter and spring, when there is a high prevalence of vitamin deficiency.[14 15] BPPV may also exhibit a seasonal pattern, with higher incidence rates seen between autumn and early spring,[16–18] although this has not been replicated in every study.[19 20]

This systematic review will examine the evidence concerning the relationship between serum vitamin D and BPPV for patients in Northern Hemisphere countries, the first time to the authors' knowledge that a review on this topic has been restricted by geographic location.

## METHODS
This systematic review was conducted following the methodology of Higgins et al[21] as closely as possible, was reported following the Preferred Reporting Items for Systematic Reviews and Meta-Analyses[22] and was preregistered in PROSPERO (CRD42021271840, see online supplemental file 1).

### Patient and public involvement
No patient involved.

### Data sources and search strategy
A systematic literature search was performed in PubMed, Scopus and Web of Science databases relating to the association between vitamin D and BPPV. Unpublished studies were not sought. Date were restricted to include studies published in English between 2000 and 2023. Three searches were run through each database to account for differing possible terms for BPPV, making up a total of 12 searches which were completed on 18 February 2023. Search criteria and results are described in online supplemental file 2, table 1. Microsoft Excel software (Microsoft Office for Windows, V.2204) was used to store and process search results.

Studies published before the year 2000 and those pertaining to animal research were removed, then any entries that did not contain keywords "BPPV", "Vertigo", "Vitamin D" or "25-hydroxyvitamin D" within their title or abstract were excluded. Entries that were not original research or did not include multiple participants (eg, case reports, letters and editorials) were also removed, leaving a total of 68 full texts to be manually reviewed. A manual search of the reference list of included full texts and systematic reviews identified by the search strategy was also performed to identify any further relevant literature. The included articles were reviewed independently by two reviewers for inclusion evaluation. Any inconsistencies between the two reviewers were evaluated further by both reviewers until a consensus was reached.

### Inclusion and exclusion criteria
To better account for seasonal differences in serum vitamin D levels, the geographical location of study participants was limited to the Northern Hemisphere. All studies that were available with English full texts that investigated the association between serum 25-hydroxyvitamin D levels in adult patients diagnosed with at least one episode of BPPV were included. Studies were excluded if participants had medical history of ear, nose and throat (ENT) comorbidities, head trauma or were taking vitamin D supplements at the start of the study.

The literature search was undertaken by HW. Study selection was independently undertaken by HW and GB, with inconsistencies resolved through discussion between the reviewers. Studies were deemed eligible for inclusion if they met the following criteria: (1) participants were over the age of 18 with at least one episode of diagnosed BPPV; (2) participants resided in countries in

**Table 1** Characteristics of included studies[24]

| Author and year | Sample size | Study design | Country | Age range | Case mean age (SD) | Case % male | Case 25(OH)D (ng/mL) mean (SD) | Quality |
|---|---|---|---|---|---|---|---|---|
| Bi et al, 2021[26] | 52 (27 case, 25 control) | Prospective case–control | China | Not reported | 51.6±15.8 | 33.33 | 14.64±6.94* | n/a RoBANS |
| Califano et al, 2019[36] | 227 (127 case, 100 control) | Retrospective+prospective case–control | Italy | Not reported | 60±11.5† | 40.2 | 24.4±10.69 | n/a RoBANS |
| Carneiro de Sousa et al, 2019[49] | 10 (5 case, 5 control) | Prospective case–control | Portugal | 52–82 | 64±9.3 (case+control) | 0 | 13.6±5.7† | n/a RoBANS |
| Çelik et al, 2021[27] | 339 (190 case, 149 control) | Prospective case–control | Turkey | Not reported | 43.7±16 | 57.4 | 15.64±8.4* | n/a RoBANS |
| Ceylan and Kanmaz, 2020[37] | 197 (97 case, 100 control) | Retrospective case–control | Turkey | Not reported | 57.07±12.6 | 34 | 15.87±8.6* | n/a RoBANS |
| Cheng et al, 2021[61] | 640 (320 case, 320 control | Prospective case–control | China | Not reported | 68.2±6.02 | 45 | 23.2±4.09 | n/a RoBANS |
| Çıkrıkçı Işık et al, 2017[38] | 127 (64 case, 63 control) | Retrospective case–control | Turkey | Not reported | 56±13.1 | 26.6 | 9.51±5.49* | n/a RoBANS |
| Ding et al, 2019[28] | 522 (174 case, 348 control) | Prospective case–control | China | Not reported | Median reported | 41.4 | 18.2 (n/a)* | n/a RoBANS |
| Elmoursey and Abbas, 2021[50] | 60 | Prospective case–control | Egypt | 24–70 | 46±12.6 | 40 | 26.1±11.7 | n/a RoBANS |
| Goldschagg et al, 2021[39] | 459 (158 cases, 301 control) | Retrospective case–control | Germany | Not reported | 61±14 | 44.9 | 23.4±9.4 | n/a RoBANS |
| Gu et al, 2018[51] | 100 (50 case, 50 control) | Prospective case–control | China | Not reported | 51.36±10.58 | 40 | 17.62±9.0 | n/a RoBANS |
| Han et al, 2018[62] | 165 (85 case, 80 control) | Retrospective case–control | China | 44–88 | 63.5±9.72 | 0 | 19.1±5.2 | n/a RoBANS |
| Han et al, 2020[29] | 117 | Retrospective cohort | Republic of Korea | 25–78 | 55±11 | 22.2 | 24.47±11.6† | Good |
| Han et al, 2021[53] | 201 | Retrospective cohort | China | Not reported | 65.14±13.35† | 0 | 19.74±6.4† | Fair |
| Inan et al, 2021[30] | 104 (52 case, 52 control) | Retrospective case–control | Turkey | 18–80 | 55.6 (n/a) | 44 | 15.3±9.8 | n/a RoBANS |
| Kahraman et al, 2016[54] | 37 | Retrospective cohort | Turkey | 23–75 | 49.8 (n/a) | 35.5 | 9.73±8.8 (initial visit)* | Good |
| Karataş et al, 2017[40] | 156 (78 case, 78 control) | Retrospective case–control | Turkey | 22–85 | 51.4±12.2 | 37.2 | 23.0±14.4 | n/a RoBANS |
| Lee et al, 2017[2] | 132 | Retrospective cohort | Republic of Korea | 49–81 | 63±10.0 | 0 | 29.43±14.2† | Fair |
| Maslovara et al, 2018[47] | 40 | Prospective cohort | Croatia | Not reported | 64.2±12.1 | 28 | 21±7.4 | Good |
| Melis et al, 2020[45] | 73 | Prospective cohort | Italy | Not reported | 62.99±14.6† | 27.4† | 22.13±15.1† | Good |
| Nakada et al, 2019[52] | 35 | Retrospective cohort | Japan | Not reported | 76.36±6.4† | 34.3† | 16.29±3.9† | Poor |
| Parham et al, 2013[43] | 29 (16 case, 13 control) | Prospective case–control | USA | 49–81 | 68.5 (n/a) | 0 | Reported as nM/L | n/a RoBANS |
| Pecci et al, 2022[41] | 50 (26 case, 24 control) | Prospective case–control | Italy | 39–79 | 61.85 (n/a) | 23 | 20.18 (SD not given) | n/a RoBANS |

Continued

**Table 1** Continued

| Author and year | Sample size | Study design | Country | Age range | Case mean age (SD) | Case % male | Case 25(OH)D (ng/mL) mean (SD) | Quality |
|---|---|---|---|---|---|---|---|---|
| Sarsitthithim et al, 2021[31] | 137 (69 case, 68 control) | Prospective case–control | Thailand | 39–89 | 61.4±11.5 | 14.5 | 21.5±5.3 | n/a RoBANS |
| Sen et al, 2018[32] | 200 (100 case, 100 control) | Prospective case–control | India | 18–60 | 48.6±10.2 | 33 | 20.3±12.2 | n/a RoBANS |
| Shu et al, 2019[16] | 877 | Retrospective cohort | China | Not reported | Median reported | Not reported | 18.03 (n/a)† | n/a RoBANS |
| Shin et al, 2023[46] | 50 | Retrospective cohort | Republic of Korea | Not reported | 50.5±14.6† | 28† | Reported by group | Good |
| Song et al, 2020[55] | 3505 (380 case, 3125 control) | Prospective case–control | China | 19–85 | 50.7±13.6† | 25.5 | 14.24±6.6† | Good |
| Talaat et al, 2015[1] | 180 (80 case, 100 control) | Prospective case–control | Kuwait | 31–71 | 47.6±9.1 | 35 | 14.19±9.3† | n/a RoBANS |
| Thomas et al, 2021[42] | 98 (49 cases, 49 controls) | Prospective case–control | India | 21–71 | 44.39 (n/a) | 32.7 | 21.3±9.567‡ | n/a RoBANS |
| Wang et al, 2020[33] | 183 (103 case, 80 control) | Prospective case–control | China | 50–86 | 63.0±9.3 | 0 | 17.2±2.0 | n/a RoBANS |
| Wu et al, 2018[34] | 152 (60 case, 92 control) | Retrospective case–control | China | Not reported | 59.35±13.24 | 100 | 23.0±6.8 | n/a RoBANS |
| Wu et al, 2022[44] | 123 (51 case; 72 control) | Prospective case–control | China | Not reported | 58.5±12.6† | 32.5 | Median reported | n/a RoBANS |
| Yang et al, 2017[48] | 260 (130 case, 130 control) | Retrospective case–control | Republic of Korea | 20–87 | 54.9±12.2 | 23.1 | 18.2±10.3 | n/a RoBANS |
| Zhang et al, 2022[35] | 206 (156 case, 50 control) | Prospective case–control | China | 46–75 | 59.5±7.4 | 0 | 18.8±2.5† | n/a RoBANS |

*Vitamin deficiency (≤20 ng/mL).
†Calculated combined average/SD combines groups of means and SD into a single group by repeating Cochrane's formula.
‡SD recalculated using reported mean and CIs.
BPPV, benign paroxysmal positional vertigo; n/a, not available; 25(OH)D, 25hydroxyvitamin D; RoBANS, Risk of Bias Assessment Tool for Nonrandomised Studies.

the Northern Hemisphere; (3) serum 25-hydroxyvitamin D levels were measured and reported and (4) participants had no medical history of head trauma or ENT comorbidities.

Articles were excluded according to the following criteria: (1) Study did not contain a reference to 'BPPV', 'vertigo' or 'vitamin D' in title or abstract; (2) Participants were taking vitamin D supplements prior to commencement of the study and (3) The article did not pertain to a research study (reviews, editorials, case studies). During application of inclusion and exclusion criteria, the decision was taken to exclude articles which included only participants with BPPV and pre-existing vitamin D deficiency as these would not be able to be utilised in subsequent analysis of the relationship between BPPV disease state and serum vitamin levels.

### Assessment of bias
Microsoft Excel and Microsoft Word software (Microsoft Office for Windows, V.2204) were used by the reviewers to record assessment of bias and document comments.

Assessment of bias was performed by two independent reviewers and any inconsistencies resolved by discussion between the reviewers. The case–control studies were assessed using the Risk of Bias Assessment Tool for Nonrandomised Studies (RoBANS)[23] which examines selection of participants, confounding variables, intervention measurement, blinding of outcome assessment, intervention measurement, incomplete outcome data and selective outcome reporting. The cohort studies were assessed according to the Newcastle-Ottawa Scale (NOS),[24] as recommended by the Cochrane Collaboration.[21] The NOS examines participant selection, comparability and outcome and assigns studies as being of 'poor', 'fair' or 'good' quality.

### Data extraction
Data extraction was completed by HW and double-checked by GB. Extracted data included study design, author, year published, location of study participants, mean age of participants, range of participant age, sample size, percentage of male participants, serum vitamin D levels for participants and the season study was performed (where reported). Standardised mean difference (SMD) with a 95% CI was used to calculate the level of vitamin D in case and control groups. Based on recommendations from the clinical practice guidelines of the Endocrine Society Task Force on vitamin D, plasma vitamin D levels of less than 20 ng/mL were considered to indicate vitamin deficiency.[25] Microsoft Excel software (Microsoft Office for Windows, V.2204) was used to tabulate the extracted data.

### Synthesis methods
Studies were included if serum 25-hydroxyvitamin D levels (mean and SD; ng/mL) were reported for cases with BPPV and controls with no medical history of BPPV. Where vitamin D levels were reported for subgroups within a study (eg, male and female means reported separately), a population mean and SD for vitamin D levels for both cases and controls was calculated according to the recommendations of the Cochrane Collaboration.[21]

The $I^2$ statistic was used to examine heterogeneity between studies, with $I^2 > 40\%$ regarded as representing significant heterogeneity. The random-effects model was to be employed for the meta-analysis if $I^2 > 40\%$, while the fixed-effect model was to be used instead if $I^2 < 40\%$. Statistical analysis was completed and forest plots were generated by using Review Manager (RevMan) software V.5.4.1. (64 bit; The Cochrane Collaboration, 2020).

## RESULTS
### Study selection
The initial searches identified 1113 results comprising 350 individual articles (figure 1). 282 of these were excluded; 163 articles were excluded because they did not contain a reference to 'BPPV' or 'vertigo' within their title or abstract, 61 were removed as they were not peer-reviewed or based on original research (review articles, conference posters, editorials, etc), 27 did not contain 'vitamin D' or related terms in their title or abstract, 6 were excluded as they related to animal studies and 25 studies were removed as they were erroneously included in search results despite being published before the year 2000.

The full text of the remaining 68 articles was reviewed for inclusion evaluation; 8 articles were excluded as the participants were receiving vitamin D supplementation before commencement of the study; 2 were excluded as they included patients with medical history of head trauma or ENT-related comorbidities; 5 were excluded as serum vitamin D levels were not measured or reported; 2 were excluded as the patients resided in countries in the Southern Hemisphere; 7 because they included participants under the age of 18 and a further 5 were excluded as they described patients with causes of vertigo other than BPPV. Four articles were also excluded as they included only participants with both BPPV and pre-existing vitamin D deficiency, precluding analysis of the relationship between disease state and serum vitamin levels. A manual search of the reference lists of these studies and identified systematic reviews did not yield any additional articles to include in the review. A total of 35 articles underwent full-length review.

## QUALITY ASSESSMENT
The 26 articles assessed with the RoBANS tool are described in online supplemental file 2, figure 1. All the studies were found to have a low risk of bias in three of the six domains assessed ('intervention measurement', 'incomplete outcome data' and 'selective outcome reporting' domains). 14 of the studies had an uncertain risk of bias in either or both of the 'selection of participant' and 'confounding variables' domains. All but two of

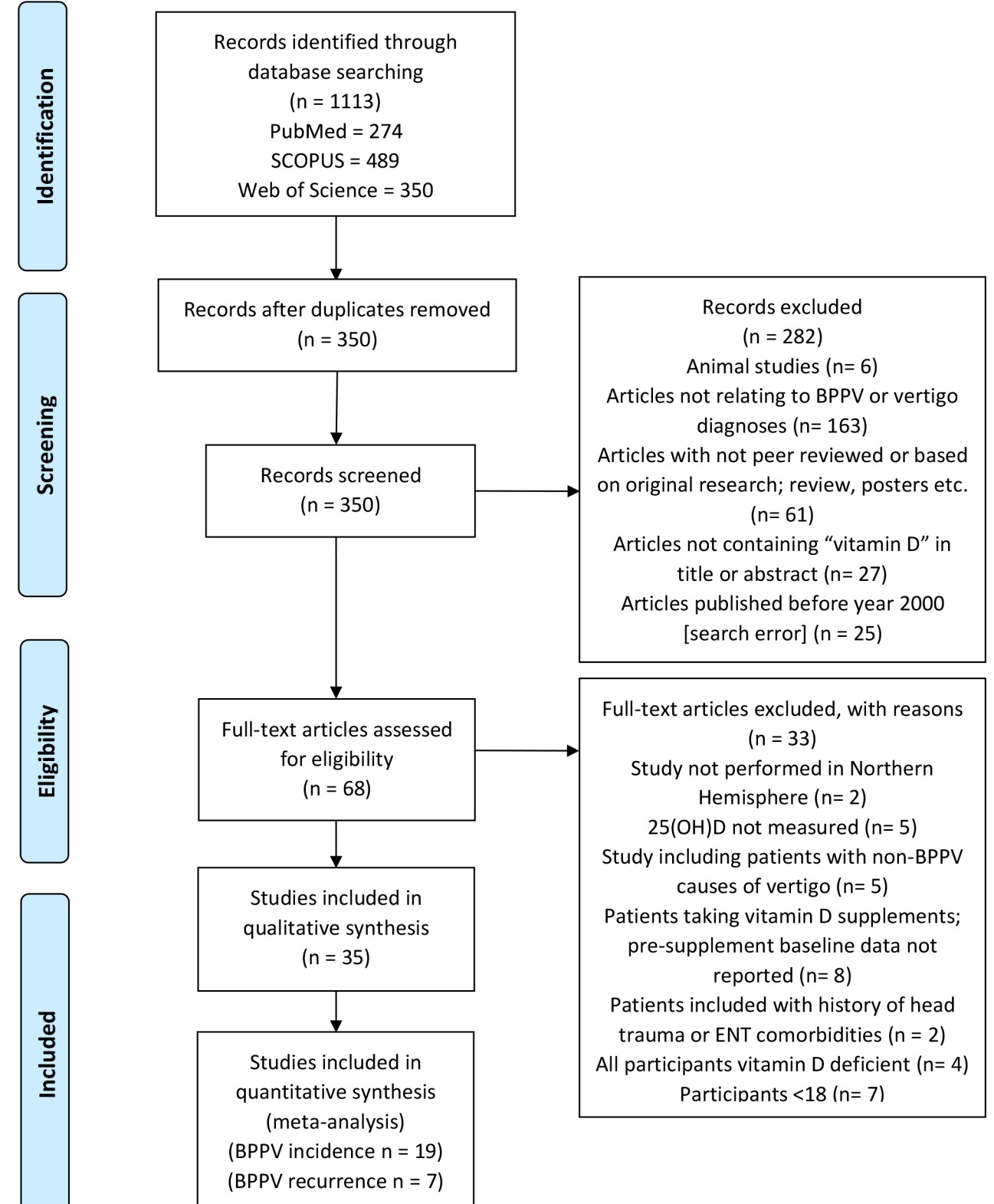

**Figure 1** Flow diagram detailing results of searches and study selection process. BPPV, benign paroxysmal positional vertigo; ENT, ear, nose and throat.

the studies had a high risk of bias in the domain relating to 'blinding of outcome assessment', four had a high risk of bias relating to 'confounding variables' and one article had a high risk of bias in the 'selection of participants'

domain. None of the articles was judged to have a low risk of bias in all six domains.

The nine cohort studies assessed using the NoS ranged in quality from 'poor' to 'good' and are described in

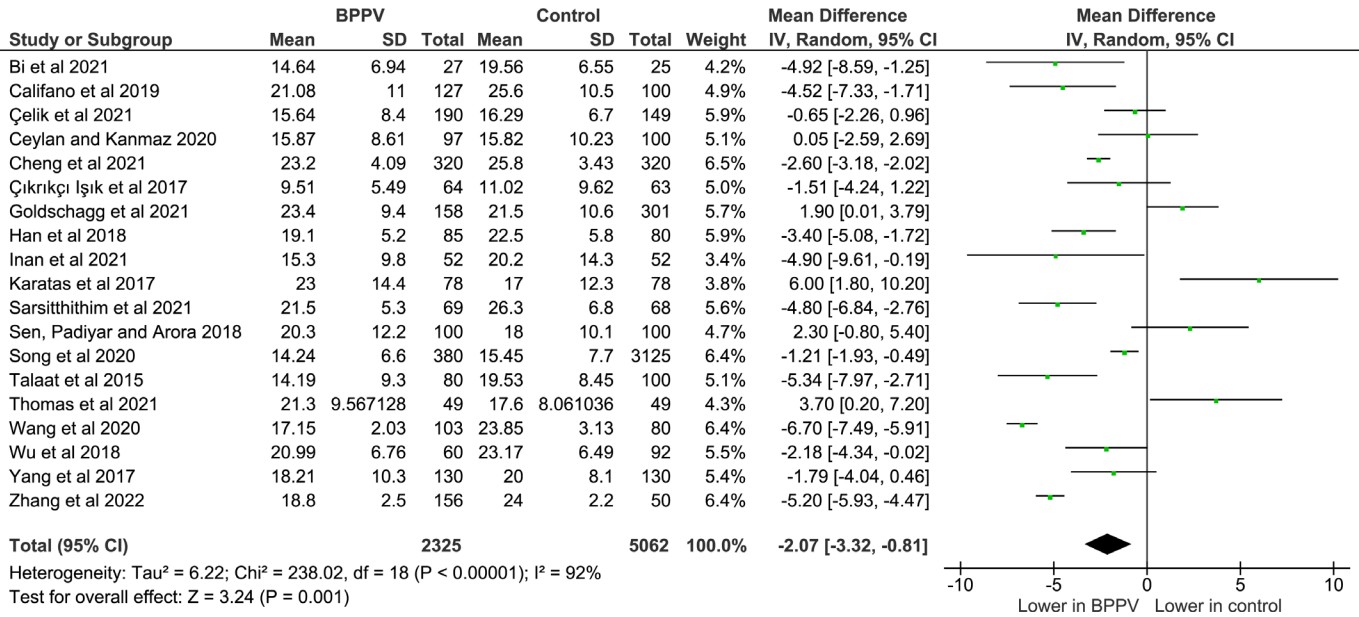

**Figure 2** Relationship between serum 25-hydroxyvitamin D and benign paroxysmal positional vertigo incidence. BPPV, benign paroxysmal positional vertigo.

online supplemental file 2, figure 2. There were significant limitations using the NoS for some of the studies, meaning that one of the eight domains of the tool ('adequacy of follow-up of cohorts') was not applicable for five of the nine studies assessed by this tool. One study was determined to be of 'poor' quality, two 'fair' quality and the remaining six studies were 'good' quality. High risk of bias was most commonly seen in the 'outcome of interest not present at start domain' and was present in seven out of nine studies. High risk of bias was seen in two studies in the 'representativeness of exposed cohort' domain and in one study for the 'follow-up long enough' domain. A single study was found to have a low risk of bias in all eight domains.

### Description of included studies

Table 1 describes the characteristics of the included studies investigating the association between serum vitamin D and BPPV for patients within Northern Hemisphere countries. The 35 studies included 9843 participants in total, of which 26 were case–control studies and 9 were cohort studies; there were no randomised controlled trials. Sample size varied from 10 to 3505 individuals, sex distribution from 0% to 100% male and age ranged between 18 and 89 years. Most studies were conducted in countries situated in Asia (12 in China, 6 in Turkey, 4 in the Republic of Korea, 2 in India and 1 each in Japan, Kuwait and Thailand); 6 studies in Europe (3 in Italy and 1 each in Croatia, Germany and Portugal); 1 study was conducted in Africa (Egypt) and 1 in North America (USA).

A total of 12 studies reported that compared with controls, patients with BPPV have lower average serum vitamin D levels.[1 26–35] Seven studies found no significant difference between the serum vitamin D levels of BPPV patients and controls.[36–42] Among cases and controls

with osteoporosis and osteopenia, two studies[2 43] found no significant difference between cases and controls, although osteoporosis was correlated with low serum vitamin D in BPPV patients.[2] Serum vitamin D levels were also reported to be lower in recovering BPPV patients with residual dizziness, compared with those without.[44]

The relationship between BPPV recurrence and serum 25-hydroxyvitamin D was examined in 12 studies, with 9 studies reporting lower serum vitamin D levels in patients with recurrent BPPV compared with those experiencing an initial attack[1 31 35–37 42 45 46] and 3 reporting no significant difference in vitamin D levels between the 2 groups.[38 47 48] Three studies[41 49 50] reported that vitamin D supplementation reduced the number of recurrent episodes of BPPV. Vitamin D supplementation was seen to improve the bone mineral density of BPPV patients in one study, however, the effects on BPPV recurrence were not reported.[51]

Whether vitamin D levels differed between subtypes of BPPV were assessed in four studies, two of which found that serum 25-hydroxyvitamin D was lower in canalithiasis compared with cupulolithiasis[47 52] while the remaining two found no significant difference in vitamin levels between BPPV subtypes.[29 53] One study reported a much higher prevalence of vitamin D deficiency (<20 ng/mL) in patients undergoing an acute attack of BPPV compared with 6 months postepisode (93.5% and 38.7%, respectively).[54]

Seasonality and BPPV were examined by two included studies; Shu et al[16] reported that the median number of BPPV referrals was higher in winter compared with summer months, while Califano et al[36] found no seasonal patterns of BPPV cases and high levels of vitamin D deficiency and insufficiency in both winter and summer.

## META-ANALYSIS

A total of 19 studies reported mean and SD for serum 25-hydroxyvitamin D levels in case (BPPV) and control (non-BPPV) groups and were included in the meta-analysis (online supplemental file 2, table 2). These described a total of 7387 participants (2325 cases; 5062 controls), ranging in age from 18 to 88 years and sex distribution from 0% to 100% male. Seven studies were excluded for the following reasons: four because the case and control groups both had BPPV,[44 46 49 51] two as SD was not reported[28 41] and one as vitamin D levels were not given in ng/mL.[43] Combined mean averages and SD were calculated for cases and controls using data provided in three studies[1 35 55] as these were initially reported according to groups (age, sex, BPPV recurrence).

SDs were recalculated for one study[42] after consultation with a statistician as reported SDs were not congruent with reported means or confidence intervals. All included studies had 'uncertain' or 'high' risk of bias in at least one domain according to RoBANs, while two[36 37] were high risk in two domains. These were not excluded as the results of both were in accordance with the rest of the studies, and neither the trend nor significance of the analysis was affected by removal of one or both studies from the analysis. Pooled SMD indicated that serum vitamin D was on average lower in cases with BPPV compared with controls (SMD −2.07; 95% CI −3.32 to −0.81; p=0.001 random effects model; figure 2). There was substantial heterogeneity across included studies ($I^2$=92%; p<0.00001).

Seven studies reported mean and SD for serum 25-hydroxyvitamin D levels in patient groups with recurrent and non-recurrent BPPV and were included in the meta-analysis (online supplemental file 2, table 3). These described a total of 622 participants (255 cases; 367 controls), ranging in age from 20 to 89 years and sex distribution from 0% to 35% male. Eight studies were excluded from the meta-analysis for the following reasons: three because no SD was reported,[28 47 56] three because serum vitamin D levels were not reported and two because vitamin D supplements were used.[36 49] All included studies were 'good' quality according to the NoS or had 'uncertain' or 'high' risk of bias in at least one domain according to RoBANs. Pooled SMD indicated that serum vitamin D is no different in individuals with recurrent BPPV compared with those with non-recurrent

disease (SMD −1.90; 95% CI −3.80 to 0.00; p=0.05 random effects model; figure 3). There was moderate heterogeneity across included studies ($I^2$=66%; p=0.007).

## DISCUSSION

Our systematic reviews and meta-analyses were based on a total of 35 articles, of which 26 case–control studies and 9 cohort studies. It provides up-to-date information regarding the evidence base for vitamin D and BPPV. The included studies described 9843 participants, of which 7510 were included in the meta-analyses. The studies included a broad range of populations from 13 different countries in the Northern Hemisphere.

Study quality was variable, and the majority of included studies were identified as having a high risk of bias in at least one domain after application of the NOS or ROBANS tools, most commonly due to a lack of blinding of outcome assessment. Some of the identified studies in this systematic review were also from journals with unclear peer-review processes or non-impact journals. Bias and potential bias were also identified relating to participant selection and unknown confounding variables, for example, participant occupation, unmatched case and control selection and incomplete data relating to relevant comorbidities including renal failure. While it was not possible to complete meta-analysis using only high-quality studies, the authors are reassured that the findings from this review are comparable to previously published literature[7 8] and did not find inclusion of studies with higher identified bias had any effect on analysis trend or significance.

The main findings of the meta-analysis were that individuals with BPPV had lower average serum 25-hydroxyvitamin D levels compared with controls, a finding which was statistically significant (p=0.001). Examining the published literature to date, it is clear that the evidence base regarding the link between vitamin D and BPPV is mixed—with a number of studies included in this review reporting no significant difference between cases with BPPV and controls and others reporting lower serum vitamin D levels in patients with BPPV. These mixed results may reflect that the strength of the relationship between vitamin D and BPPV is relatively weak. Another possibility is that rather than vitamin D having

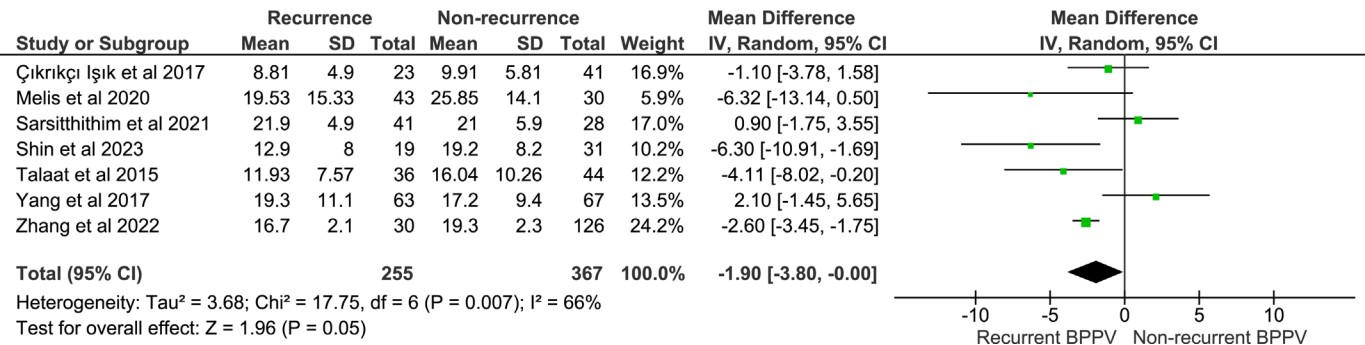

**Figure 3** Relationship between serum 25-hydroxyvitamin D and benign paroxysmal positional vertigo recurrence.

a direct physiological role in the development of BPPV it is an indirect marker of another related biomarker, for example, calcium; as calcitriol, manufactured from 25-hydroxyvitamin D, plays a role in regulating intestinal absorption of dietary calcium and bone mineralisation both directly and through interaction with parathyroid hormone.[57]

The effect size in our meta-analysis was relatively small (SMD −2.07), which aligns with other non-geographically restricted systematic reviews performed previously which have established an association between low serum vitamin D and BPPV incidence.[8 9] It remains unclear, however, whether the small absolute difference in serum vitamin D concentration reported between patients with BPPV and those without is physiologically relevant.

The results of our meta-analysis into the association between BPPV recurrence and serum vitamin D levels indicated a similar (SMD −1.90), although non-significant association between BPPV recurrence and lower serum vitamin D levels. Previously published, non-geographically restricted systematic reviews have demonstrated a mixed picture relating to the relationship between serum vitamin D and BPPV recurrence with both a negative association,[8 10] or no impact on recurrence[11] reported.

It might have been expected that restricting our review to include only studies conducted in the Northern Hemisphere would result in a larger effect size, as more northerly latitudes have previously been shown to be a risk factor for vitamin D deficiency.[58 59] However, as vitamin D deficiency is more prevalent in such countries during the winter,[60] this may have been confounded by the fact that most of the included studies did not include temporal data relating to the month and/or seasons when vitamin D levels were measured or BPPV occurred. Some of the articles included in this review indicated mixed evidence for a potential relationship between serum vitamin D levels and subtypes of BPPV. Two articles[47 52] found lower serum vitamin D levels in canalithiasis compared with cupulolithiasis, while a further two found no significant difference in vitamin D levels between the BPPV subtypes.[29 53] It was not possible to perform a meta-analysis on this effect due to the small number of studies investigating this relationship, however, this could be a parameter for future studies investigating BPPV and vitamin D to explore further. Similarly, the potential association between BPPV, vitamin D and bone mineral density identified by some of the articles[2 43] could not be further analysed with meta-analysis but could be an area of interest for future research.

It was not possible to perform a meta-analysis examining the seasonality of both BPPV and vitamin D as only two studies reported vitamin D levels according to month or season.[16 36] This is an important potential confounding factor, as vitamin D levels have been observed to be at their lowest during months when the incidence of BPPV is greatest.[18] One hypothesis is that higher levels of vitamin D may offer a protective effect against BPPV during the summer months and that the decrease in vitamin D levels over the winter months results in the increased incidence of BPPV seen during these months. Future research should separate individuals by month or season of admission (summer and winter-spring) and examine the difference in serum vitamin D levels between patients and controls during these time periods.

In conclusion, our results indicate that low serum vitamin D levels are correlated with the incidence of BPPV, while a non-significant association was reported in recurrent BPPV compared with non-recurrent disease. It was not possible to further investigate the seasonal relationships of BPPV and vitamin D deficiency as most studies do not include data relating to when patients are seen, and this is an area of interest for future research. Investigating if serum vitamin D levels relate to subtypes of BPPV and whether restoring vitamin D levels to sufficient levels via supplementation offers protection against BPPV are other topics which would benefit from further studies. As a large proportion of the population of countries in Northern Hemisphere countries are at a greater risk of vitamin D deficiency, patients developing BPPV in these countries may benefit from opportunistic serum vitamin D testing and/or supplementation.

**Contributors** HW, KK and GB conceptualised, planned and designed this systematic review and meta-analyses. HW conducted the searches and initial screening of titles and abstracts. HW and GB independently reviewed the full texts and conducted the assessment of bias. HW extracted the data, and this was verified by GB, disagreements at any stage were resolved by discussion and by involvement of KK. HW and GB interpreted the results, and this was reviewed by KK. HW wrote the original manuscript. GB and KK were responsible for supervision, administration, reviewing and editing the manuscript. All authors reviewed the results and approved the final version of the manuscript. GB and KK are responsible foroverall content as the guarantors.

**Funding** KK was supported by the NIHR Manchester Biomedical Research Centre, UK IS-BRC-1215-20007, NIHR203308 and MRC Programme Grants MR/L003589/1, MR/V01272X/1.

**Competing interests** None declared.

**Patient and public involvement** Patients and/or the public were not involved in the design, or conduct, or reporting, or dissemination plans of this research.

**Patient consent for publication** Not applicable.

**Ethics approval** Ethics approval was not necessary for this study as no new data were collected and no patient-identifiable data included.

**Provenance and peer review** Not commissioned; externally peer reviewed.

**Data availability statement** All data relevant to the study are included in the article or uploaded as online supplemental information.

**ORCID iDs**
Karolina Kluk http://orcid.org/0000-0003-3638-2787

Ghada BinKhamis http://orcid.org/0000-0002-7696-1073

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
