## [Reviewer comments · BMJ Open]

ARTICLE DETAILS

TITLE (PROVISIONAL)	Association between vitamin D deficiency and Benign Paroxysmal Positional Vertigo (BPPV) incidence and recurrence: A systematic review and meta-analysis
AUTHORS	Wood, Heather; Kluk, Karolina; Binkhamis, Ghada

VERSION 1 – REVIEW

REVIEWER	Casani, Augusto Pietro Department of Neuroscience, Otolaryngology Section, Pisa University Hospital
REVIEW RETURNED	07-Aug-2023

GENERAL COMMENTS	this systematic review appears to be conducted with extreme metaodological and statistical rigour; in particular, it seemed to me very correct to indicate in the conclusions the insufficient evidence on the role of vitamin D on the incidence and recurrence of BPPV. In particular, it seemed to me extremely important to indicate the absence of data that could correlate seasonality with the incidence of BPPV in relation to any vitamin D deficiency. I fully agree with the results of the rigorous systematic review on the need for studies evaluating all comorbidities and all other factors (age, gender, country of origin and residence of patients) in order to evaluate the effective role of vitamin D in genesis of BPPV and its recurrence. Just one consideration: some studies considered for the systematic review are published in journals in which the existence of a peer reviewed process does not appear clear (for example Califano et al, 2019, for which the bibliographic reference appears incomplete) or in non-impact journals and therefore scientific value at least questionable (e.g. Ceylan et al). I think it is correct, if not to eliminate them from the review, at least to underline this aspect.
--

REVIEWER	Chauhan, Ishan IGMC SHIMLA, ENT - HEAD AND NECK SURGERY
REVIEW RETURNED	11-Aug-2023

GENERAL COMMENTS	Overall a good systematic review..few suggestions: 1. This study has limitation regarding occupational data of the subjects which can vary both vitamin d deficiency as well as BPPV. This should be added to limitations.2. DHI scoring to grade severity of BPPV could have been added to better assess role of vitamin d in it's treatment.3. More studies could have been included to study relapse/ recurrence in BPPV with vitamin d deficiency.
---

	4. In line 36 BPPV is written as BBPV can be corrected also language correction advised in first para of synthesis method.
--	--

REVIEWER	Posey, Rachael Venebio Group LLC
REVIEW RETURNED	28-Aug-2023

GENERAL COMMENTS	Abstract: If possible, it would be helpful to see a more thorough listing of the limitations of this study in the limitations section of the abstract, though this may be limited by word count. Examples include limitations of the systematic review methodology itself (e.g., a SR is inherently limited by its retrospective design), as well as a mention of the limitations imposed by the inclusion/exclusion criteria, as mentioned in the Discussion section (e.g., limiting to the northern hemisphere may have biased results), and limitations imposed by the available studies (e.g., overall variable study quality, publication/non-reporting bias, lack of available RCTs) Methods: It is not clear why the searches were limited to publications published in the year 2000 or later. The authors should provide a justification for this decision and potentially discuss this as a limitation of the findings if it is possible that there may be relevant studies published prior to 2000. There are errors and omissions in the search strategies that make it impossible to determine if all relevant studies were identified. For example, the term "Vitamin D*" cannot be used in PubMed, as PubMed will not allow truncation for words of three or fewer characters, and therefore, this term did not run as it appears to have been intended. As such, this search did not pick up the "Vitamin D" MeSH term, which would seem to be highly relevant to this topic. The search is also missing field tags, plurals (e.g., vitamins, vertigos), and synonyms that may be relevant (the MeSH database suggests that "Familial Vestibulopathy" and "Benign Recurrent Vertigo" may be synonyms for BPPV that are not represented here). Without a systematic search strategy, it is impossible to determine if all relevant articles were identified and, as such, if the conclusions drawn here are valid, as missing studies may have affected the results of the meta-analysis as well as the qualitative synthesis of the studies. Both the Cochrane Handbook and the IoM standards recommend consulting with an experienced medical/healthcare librarian or information specialist in the development of the search strategies (https://training.cochrane.org/handbook/current/chapter-04; https://nap.nationalacademies.org/catalog/13059/finding-what-works-in-health-care-standards-for-systematic-reviews). No librarian appears to have been consulted. Given the flaws in the search strategy, the authors should consult with a librarian prior to resubmission or explain why this step is not possible. The Higgins citation mentioned in the methods section is for the Cochrane randomized trial assessment tool. Given the placement of the citation and fact that no randomized trials were included in this review, I assume that this should point to the Cochrane Handbook (https://training.cochrane.org/handbook/current) rather than the RoB tool.
---

The rationale for restricting the review to the northern hemisphere is not entirely clear. If the intention is to examine the effect of seasonality, then one would assume countries at relative distances from the equator would be similarly affected. Further explanation is needed to justify the exclusion of, e.g., New Zealand or Chile but not Egypt or India. The authors should provide further justification for this decision in order for the validity of their hypothesis to be assessable.

Results:

Study quality was assessed, but it is unclear how these ratings were used in the evidence synthesis. Was study quality considered in the meta-analyses or in the qualitative synthesis (i.e., write-up/discussion)? Were studies deemed to be of better quality weighted more strongly than studies subject to higher risk of bias or were there differences in the reported results for studies that were subject to higher risk of bias versus those with fewer potential biases? How may the specific sources of bias identified have affected the reported results? The authors should discuss if and how potential biases affected the synthesis of results in both the results and discussion sections (<https://training.cochrane.org/handbook/current/chapter-07#section-7-6>).

The word “experimental” appears to be used in error here. The included studies (case-control and cohort studies) were not experimental in design, and therefore, studies could not have been excluded due to not being experimental. Based on the specific examples cited (case reports, review articles, conference posters, editorials), it appears that publications were excluded if they were not based on original research, were not peer reviewed, or were based on isolated case reports rather than due to their lack of experimental design. The authors should clarify this.

A manual search of the reference lists of studies reviewed at the full-text level was performed, but it is not clear if the reference lists of other relevant systematic reviews were also manually reviewed. The authors should clearly state if this step was conducted, as it is recommended by the Cochrane Handbook (<https://training.cochrane.org/handbook/current/chapter-04#section-4-3>).

The authors state that conference papers were excluded, and no other mention of unpublished/grey literature is made. As such, it appears that unpublished data were not searched for or assessed. Without an assessment of unpublished literature and/or a funnel plot analysis, it is impossible to determine whether or not this literature base may be subject to publication/non-reporting bias (<https://training.cochrane.org/handbook/current/chapter-04#section-4-3>; <https://training.cochrane.org/handbook/current/chapter-13>). The authors should clearly state if non-published (grey) literature was excluded and if so, why. The lack of a publication bias analysis should also be mentioned as a limitation of the present review.

In the paragraph describing which studies were included/excluded from meta-analysis, it would be helpful to include reference numbers so that the reader can ascertain which studies were excluded from the meta-analysis for which reason.

Given the high degree of heterogeneity, it would be useful if the authors provided an explanation or attempted explanation for this heterogeneity (e.g., did populations or methods of data collection differ significantly between studies). The high degree of heterogeneity may also mean that substantial differences in the methods and populations assessed in these studies may make this body of literature unsuitable for meta-analysis or for the specific meta-analyses conducted here.

There appears to be a mistake in the 95% CI reported for the second meta-analysis. Figure 3 reports the 95% CI as -3.85 to 0.02, while the text reports the 95% CI as -3.85 to -0.02). If the upper bound does not actually cross zero, then that should mean that this meta-analysis did find a significant difference between groups, which does not appear to be the case based on the surrounding text and the figure. The authors should assess which is correct and, if necessary, adjust the text accordingly.

Discussion:

The discussion states that 8002 individuals were included in the meta-analysis, while the abstract states that there were 7387 individuals in one meta-analysis and 615 in the other. As some studies were included in both of the analyses, it seems reasonable to assume that there would be some overlap between the individuals in both analyses, and as such, the total number of included individuals would be expected to be less than the sum of individuals in each of the two meta-analyses ($615+7387=8002$).

The two cited publications indicating that Vitamin D deficiency is associated with more northerly latitudes compare populations within the northern hemisphere against each other (e.g., people living at higher latitudes within the northern hemisphere seem to have higher rates of vitamin D deficiency than those living at lower latitudes within the northern hemisphere). This review does not appear to be comparing populations at different latitudes within the same hemisphere, but rather it groups all populations within the northern hemisphere and posits that a stronger effect should be seen in this population than in analyses that examine populations in both northern and southern hemispheres. Without a comparison between groups at different latitudes or the exclusion of countries below a specific latitude within the northern hemisphere, it does not appear that the effect seen here could be or should be expected to be different from studies that looked at the global population. As such, the rationale for limiting included studies only to those conducted in the northern hemisphere is lacking.

Given the flaws in the search strategies, the lack of a manual review of the publications cited by other systematic reviews, and the lack of an analysis of publication/non-reporting bias, it is difficult to determine if the conclusions reached here are valid. Missing studies may have biased the results in one direction or another. The high degree of heterogeneity seen here also casts doubt on the appropriateness of the meta-analyses conducted here, as it seems doubtful that the differences in results seen here are due entirely to change and not to other underlying differences between studies, and as such, the results of the meta-analyses may not be particularly meaningful (<https://training.cochrane.org/handbook/current/chapter-10#section-10-10>).

REVIEWER	Sharif, Sameer McMaster University
REVIEW RETURNED	05-Nov-2023

GENERAL COMMENTS	Thank you for the opportunity to review this systematic review and meta-analysis on the association between Vitamin D deficiency and BPPV incidence and recurrence. I was asked to do a statistical review of this study. Please see below for my feedback as an opportunity to further improve upon your work. Methods Search:  - Why were only studies since 2000 included? Why not earlier ones? - I see the logic of why only participants in the Northern Hemisphere included in the analysis? - - It would be interesting to include both Northern and Southern hemisphere patients and then do a subsequent subgroup analysis - Was a PRISMA form completed? Could you complete one and include it in the Appendix for review? - I believe RoBANS still requires validation? Did you consider using the ROBINS tool? Cochrane recommends the use of the ROBINS tool. Random Effects vs Fixed Effects  - Random effects should be employed here despite what the I2 is. Fixed effects does not simply relate to statistical heterogeneity but also as to how the studies were conducted. If the Methods employed by the study were incredibly similar, an argument can be made to use the Fixed-Effects model. This is not the case here. GRADE: I do not see that this was utilized  - The Methodological standard for reporting meta-analyses is to use GRADE. Please go on the GRADE website <https://www.gradepro.org/> and create a summary of evidence table for free. This will allow readers and reviewers to determine the certainty of the evidence. Thank you once again for allowing me to review your work. I look forward to the opportunity to review it once again after the above suggestions have been accounted for. Specifically, the major things that need revision is the use of the ROBINS tool and using GRADE.
---

REVIEWER	Raj, Dharma ICMR
REVIEW RETURNED	20-Nov-2023

GENERAL COMMENTS	No comments.
--------------

VERSION 1 – AUTHOR RESPONSE

Reviewer 1:

Dr. Augusto Pietro Casani, Department of Neuroscience, Otolaryngology Section, Pisa University

Hospital

Comments to the Author:

This systematic review appears to be conducted with extreme methodological and statistical rigour; in particular, it seemed to me very correct to indicate in the conclusions the insufficient evidence on the role of vitamin D on the incidence and recurrence of BPPV. In particular, it seemed to me extremely

important to indicate the absence of data that could correlate seasonality with the incidence of BPPV in relation to any vitamin D deficiency.

I fully agree with the results of the rigorous systematic review on the need for studies evaluating all comorbidities and all other factors (age, gender, country of origin and residence of patients) in order to evaluate the effective role of vitamin D in genesis of BPPV and its recurrence.

Thank you for your positive feedback.

- Just one consideration: some studies considered for the systematic review are published in journals in which the existence of a peer reviewed process does not appear clear (for example Califano et al, 2019, for which the bibliographic reference appears incomplete) or in nonimpact journals and therefore scientific value at least questionable (e.g. Ceylan et al). I think

it is correct, if not to eliminate them from the review, at least to underline this aspect

- o Many thanks. The authors agree that high-quality research is currently limited in this topic. However, none of the tools utilised to assess risk of bias include specific

consideration of journal impact value or source, so this was not addressed as part of

our methodology. Additionally, the protocol for the review was pre-registered and

published in PROSPERO prior to undertaking the work, and exclusion of work based

on lack of a clear peer review process or being in a non-impact journal was not a predefined exclusion criterion.

- o A comment relating to the limited availability of high-quality research and the

questionable peer review processes has been added to the limitations at the start of

the revised manuscript and the discussion as follows:

- Limitations (Page 3): "High quality research is currently limited, and a few of

the identified studies in this systematic review were from journals with an

unclear peer review processes or non-impact journals.”

▪ Discussion (Page 11): “. Some of the identified studies in this systematic review were also from journals with unclear peer review processes or non-impact journals.”

Reviewer 2:

Dr. Ishan Chauhan, IGMC SHIMLA

Comments to the Author:

Overall a good systematic review... few suggestions:

Thank you for this positive comment.

• This study has limitation regarding occupational data of the subjects which can vary both vitamin D deficiency as well as BPPV. This should be added to limitations.

o Many thanks. The following statement has been added to the limitations section of the revised manuscript to address this (page 3): “Data were not available relating to factors potentially related to both BPPV and vitamin D such as subject occupation, comorbidities and symptom severity.”

• DHI scoring to grade severity of BPPV could have been added to better assess role of vitamin D in its treatment.

o The authors agree that investigating a relationship between vitamin D levels and the severity of BPPV symptoms would have been interesting to characterise. However, this was not possible as only one of the papers identified by the review (Wu et al 2022) included data relating to the dizziness handicap index (DHI).

o This statement was added to the limitations section of the revised manuscript (page 3): “Data were not available relating to factors potentially related to both BPPV and vitamin D such as subject occupation, comorbidities and symptom severity”.

• More studies could have been included to study relapse/ recurrence in BPPV with vitamin d deficiency.

o All studies identified in the literature search relating to BPPV relapse/recurrence were

included in both the systematic review and meta-analysis following application of inclusion and exclusion criteria. The protocol for the systematic review was pre-registered and published in PROSPERO (CRD42021271840) and the review was completed according to this protocol.

- In line 36 BPPV is written as BBPV can be corrected also language correction advised in first para of synthesis method.

- o Many thanks, this error has now been amended.

Reviewer 3:

Ms. Rachael Posey, Venebio Group LLC

Comments to the Author:

Abstract:

- If possible, it would be helpful to see a more thorough listing of the limitations of this study in the limitations section of the abstract, though this may be limited by word count. Examples include limitations of the systematic review methodology itself (e.g., a SR is inherently limited by its retrospective design), as well as a mention of the limitations imposed by the inclusion/exclusion criteria, as mentioned in the Discussion section (e.g., limiting to the northern hemisphere may have biased results), and limitations imposed by the available studies (e.g., overall variable study quality, publication/non-reporting bias, lack of available RCTs):

- o Many thanks, whilst space is indeed limited, further acknowledgement of limitations has been added relating to the limited availability of high-quality research and confounding factors. The following statements were added to the limitations section in the revised manuscript (page 3):

- “Data were not available relating to factors potentially related to both BPPV and vitamin D such as subject occupation, comorbidities and symptom severity.”
- “High-quality research is currently limited, and a few of the identified studies

in this systematic review are from journals with unclear peer-review processes or non-impact journals.”

Methods:

- It is not clear why the searches were limited to publications published in the year 2000 or later. The authors should provide a justification for this decision and potentially discuss this as a limitation of the findings if it is possible that there may be relevant studies published prior to 2000.

- o Searches were limited to ensure that included research was reasonably up to date.

The Systematic Review protocol including this search strategy were pre-registered and published in PROSPERO prior to undertaking the systematic review.

- There are errors and omissions in the search strategies that make it impossible to determine if all relevant studies were identified. For example, the term “Vitamin D*” cannot be used in PubMed, as PubMed will not allow truncation for words of three or fewer characters, and therefore, this term did not run as it appears to have been intended. As such, this search did not pick up the “Vitamin D” MeSH term, which would seem to be highly relevant to this topic. The search is also missing field tags, plurals (e.g., vitamins, vertigos), and synonyms that may be relevant (the MeSH database suggests that “Familial Vestibulopathy” and “Benign Recurrent Vertigo” may be synonyms for BPPV that are not represented here). Without a systematic search strategy, it is impossible to determine if all relevant articles were identified and, as such, if the conclusions drawn here are valid, as missing studies may have affected the results of the meta-analysis as well as the qualitative synthesis of the studies.

- o Thank you for your comments, prior to undertaking this systematic review, our protocol including the search strategy was pre-registered and published in PROSPERO.

- o Both “Benign Recurrent Vertigo” and “familial vestibulopathy” (also known as bilateral vestibulopathy and sometimes abbreviated to “BVP”) are diagnoses that are clinically distinct from benign paroxysmal positional vertigo (BPPV) and are not within the scope of this systematic review.

- o The error stating that the search term “vitamin D*” was used in PUBMED has been amended as it should have been “vitamin D” only, as it was for Web of Science, thank you for pointing this out. The authors are reassured that the concurrent search terms (25-hydroxyvitamin D, ergocalciferol, cholecalciferol) will have detected the required relevant results as the D in “vitamin D” is not an abbreviation or truncation.
- Both the Cochrane Handbook and the IoM standards recommend consulting with an experienced medical/healthcare librarian or information specialist in the development of the search strategies (<https://training.cochrane.org/handbook/current/chapter-04>; <https://nap.nationalacademies.org/catalog/13059/finding-what-works-in-health-carestandards-for-systematic-reviews>). No librarian appears to have been consulted. Given the flaws in the search strategy, the authors should consult with a librarian prior to resubmission or explain why this step is not possible.
- o Many thanks for your comments, the authors hope that the response to the previous comment will reassure the reviewer that there are not significant flaws present in the current search strategy.
- o One of the authors involved in this systematic review has extensive experience in systematic reviews and meta-analysis. They have worked extensively with librarians prior to undertaking this review and were able to utilise their prior experience when formulating the search strategy.
- o Additionally, the search strategy was pre-registered, reviewed, approved and published as part of a PROSPERO protocol prior to work being undertaken.
- The Higgins citation mentioned in the methods section is for the Cochrane randomized trial assessment tool. Given the placement of the citation and fact that no randomized trials were included in this review, I assume that this should point to the Cochrane Handbook (<https://training.cochrane.org/handbook/current>) rather than the RoB tool.
- o Many thanks, this referencing error has now been amended.
- The rationale for restricting the review to the northern hemisphere is not entirely clear. If the intention is to examine the effect of seasonality, then one would assume countries at relative

distances from the equator would be similarly affected. Further explanation is needed to justify the exclusion of, e.g., New Zealand or Chile but not Egypt or India. The authors should provide further justification for this decision in order for the validity of their hypothesis to be assessable.

o Before commencing, one of the aims of this review was to investigate possible seasonal relationships between both vitamin D and BPPV. Current literature focussed on countries within the Northern hemisphere has demonstrated that serum vitamin D levels vary according to season, which is why this review restricted searches to only countries within the Northern hemisphere.

Results:

- Study quality was assessed, but it is unclear how these ratings were used in the evidence synthesis. Was study quality considered in the meta-analyses or in the qualitative synthesis (i.e., write-up/discussion)? Were studies deemed to be of better quality weighted more strongly than studies subject to higher risk of bias or were there differences in the reported results for studies that were subject to higher risk of bias versus those with fewer potential biases? How may the specific sources of bias identified have affected the reported results?

The authors should discuss if and how potential biases affected the synthesis of results in both the results and discussion sections

(<https://training.cochrane.org/handbook/current/chapter-07#section-7-6>).

o Many thanks, the results and discussion have been updated accordingly.

o Of the studies assessed by the NoS, one was study deemed to be of “poor” quality and two of “fair” quality. None of these studies fit the criteria to be included in the subsequent meta-analyses.

o Of the studies assessed using the RoBANS tool, most scored high risk of bias for lack of blinding of outcome assessment. However, as the outcome being measured (a serum vitamin D measurement) is objective, the lack of blinding would not be able to influence blood serum measurement. For this reason, studies were not excluded from

meta-analysis on this basis.

o The pre-registered protocol available on PROSPERO did not include criteria to exclude studies from meta-analysis based on level of bias.

o As many studies were judged to have bias defined as “uncertain” or “poor” based on unknown potential confounding variables, it was not possible to weight or quantify bias in studies in the meta-analysis based on unknown variables.

o Statements addressing this were added in the results and discussion sections of the revised manuscript as follows:

- Results section (Page 8): “All included studies had ‘uncertain’ or ‘high’ risk of bias in at least one domain according to RoBANS, whilst two³⁶ ³⁷were high risk in two domains.

These were not excluded as the results of both were in accordance with the rest of the studies, and neither the trend or significance of the analysis was affected on removal of one or both of the studies from the analysis.”

- Discussion section (Page 11): “Bias and potential bias was also identified relating to participant selection and unknown confounding variables, for example participant occupation, unmatched case and control selection and incomplete data relating to relevant comorbidities including renal failure. Whilst it was not possible to complete meta-analysis using only high-quality studies, the authors are reassured that the findings from this review are comparable to previously published literature^{7 8}, and did not find inclusion of studies with higher identified bias had any effect on analysis trend or significance.”

• The word “experimental” appears to be used in error here. The included studies (case-control and cohort studies) were not experimental in design, and therefore, studies could not have been excluded due to not being experimental. Based on the specific examples cited (case reports, review articles, conference posters, editorials), it appears that publications were excluded if they were not based on original research, were not peer reviewed, or were based on isolated case reports rather than due to their lack of experimental design. The authors

should clarify this.

o Many thanks for your suggestion. The manuscript has been reworded in the methods and results sections of the revised manuscript for improved clarity relating to excluded studies as follows:

- Methods (Page 5): “Entries that were not original research or did not include multiple participants e.g., case reports, letters and editorials were also removed, leaving a total of 68 full texts to be manually reviewed.”
- Results (Page 6): “61 were removed as they were not peer reviewed or based on original research (review articles, conference posters, editorials etc.),”
- A manual search of the reference lists of studies reviewed at the full-text level was performed, but it is not clear if the reference lists of other relevant systematic reviews were also manually reviewed. The authors should clearly state if this step was conducted, as it is recommended by the Cochrane Handbook (<https://training.cochrane.org/handbook/current/chapter04#section-4-3>).

o Many thanks. A manual search of reference lists was completed and referred to at the end of the second paragraph under the results subheading “study selection” in the revised manuscript as follows (Page 7): “A manual search of the reference lists of these studies and identified systematic reviews did not yield any additional articles to include in the review. A total of 35 articles underwent full-length review.”

o An additional sentence has been added to the “data sources and search strategy” section of the Methods in the revised manuscript to reflect that manual search of reference lists took place as follows (Page 5): “A manual search of the reference list of included full texts and systematic reviews identified by the search strategy was also performed to identify any further relevant literature.”

- The authors state that conference papers were excluded, and no other mention of unpublished/grey literature is made. As such, it appears that unpublished data were not searched for or assessed. Without an assessment of unpublished literature and/or a funnel plot analysis, it is impossible to determine whether or not this literature base may be subject

to publication/non-reporting bias (<https://training.cochrane.org/handbook/current/chapter04#section-4-3>; <https://training.cochrane.org/handbook/current/chapter-13>). The authors

should clearly state if non-published (grey) literature was excluded and if so, why. The lack of a publication bias analysis should also be mentioned as a limitation of the present review.

o This search strategy, including exclusion of unpublished studies were pre-registered and published in PROSPERO prior to undertaking the systematic review. The Cochrane handbook states that searching relevant grey literature is “highly desirable” but not “mandatory” (<https://training.cochrane.org/handbook/current/chapter-04>).

o A sentence “Unpublished studies were not sought.” has been added to the methods of the revised manuscript under the subheading “Data sources and search strategy” (Page 5) to reflect this.

• In the paragraph describing which studies were included/excluded from meta-analysis, it would be helpful to include reference numbers so that the reader can ascertain which studies were excluded from the meta-analysis for which reason.

o Many thanks for this suggestion, references have now been incorporated to indicate which studies were excluded for which reason in the meta-analysis.

• Given the high degree of heterogeneity, it would be useful if the authors provided an explanation or attempted explanation for this heterogeneity (e.g., did populations or methods of data collection differ significantly between studies). The high degree of heterogeneity may also mean that substantial differences in the methods and populations assessed in these studies may make this body of literature unsuitable for meta-analysis or for the specific metaanalyses conducted here.

o Heterogeneity was used to help to determine the appropriate statistical test to use for meta-analysis (fixed effects vs random effects model), and the values reported were congruent with previously published meta-analyses investigating vitamin D levels and BPPV, which range from moderate to high, e.g.,:

▪ Yang, B., Lu, Y., Xing, D., Zhong, W., Tang, Q., Liu, J., & Yang, X. (2020).

Association between serum vitamin D levels and benign paroxysmal positional vertigo: a systematic review and meta-analysis of observational

studies. *European Archives of Oto-Rhino-Laryngology*, 277, 169-177.

<https://link.springer.com/article/10.1007/s00405-019-05694-0>

▪ AlGarni, M. A., Mirza, A. A., Althobaiti, A. A., Al-Nemari, H. H., & Bakhsh, L. S.

(2018). Association of benign paroxysmal positional vertigo with vitamin D

deficiency: a systematic review and meta-analysis. *European Archives of OtoRhino-Laryngology*, 275, 2705-2711.

<https://link.springer.com/article/10.1007/s00405-018-5146-6>

o Input was received from two independent statisticians (reviewer four and a statistician at the University of Manchester) regarding the meta-analyses and advice was given relating to the most appropriate model to use (fixed vs random effects) based on heterogeneity. Meta-analyses were repeated following statistician advice and figures/values have been updated accordingly.

o As the measures analysed were a standardised blood test for serum vitamin D, and diagnosis of BPPV (a common and well characterized disease), it is unlikely that the observed differences between studies would make the data unsuitable for metaanalysis. Heterogeneity may reflect differences between population vitamin D levels between countries, but this would not invalidate examination of vitamin D levels and BPPV incidence between studies.

o Additionally, the values reported were congruent with previously published metaanalyses investigating vitamin D levels and BPPV, which range from moderate to high,

e.g.,:

▪ Yang, B., Lu, Y., Xing, D., Zhong, W., Tang, Q., Liu, J., & Yang, X. (2020).

Association between serum vitamin D levels and benign paroxysmal positional vertigo: a systematic review and meta-analysis of observational studies. *European Archives of Oto-Rhino-Laryngology*, 277, 169-177.

<https://link.springer.com/article/10.1007/s00405-019-05694-0>

▪ AlGarni, M. A., Mirza, A. A., Althobaiti, A. A., Al-Nemari, H. H., & Bakhsh, L. S.

(2018). Association of benign paroxysmal positional vertigo with vitamin D

deficiency: a systematic review and meta-analysis. *European Archives of OtoRhino-Laryngology*, 275, 2705-2711.

<https://link.springer.com/article/10.1007/s00405-018-5146-6>

- There appears to be a mistake in the 95% CI reported for the second meta-analysis. Figure 3 reports the 95% CI as -3.85 to 0.02, while the text reports the 95% CI as -3.85 to -0.02). If the upper bound does not actually cross zero, then that should mean that this meta-analysis did find a significant difference between groups, which does not appear to be the case based on the surrounding text and the figure. The authors should assess which is correct and, if necessary, adjust the text accordingly.

- o Thank you for drawing our attention to this error. This meta-analysis has been repeated after consultation with a statistician, and values have been updated to reflect this.

Discussion:

- The discussion states that 8002 individuals were included in the meta-analysis, while the abstract states that there were 7387 individuals in one meta-analysis and 615 in the other. As some studies were included in both of the analyses, it seems reasonable to assume that there would be some overlap between the individuals in both analyses, and as such, the total number of included individuals would be expected to be less than the sum of individuals in each of the two meta-analyses ($615+7387=8002$).

- o Many thanks for pointing out this error. There were 19 papers included in the metaanalysis for vitamin D levels in BPPV incidence vs. controls and seven papers included

in the meta-analysis for BPPV recurrence vs non-recurrence. Four papers are included in both meta-analyses (Çıkrıkçı Işık et al. 2017; Sarsithithim et al. 2021; Talaat et al. 2015; Yang et al. 2017 and Zhang et al. 2022).

- o The reviewer is correct that there is some overlap in the included individuals. This is now reflected in the new total (7510).

- o Please also be advised of a slight adjustment in the recurrence meta-analysis following further statistical input, which has resulted in the number of cases being included in the recurrence meta-analysis from the Çıkrıkçı Işık et al. 2017 study increasing from 16 to 23. Totals for the recurrence meta-analysis, and new totals have been accounted

for.

- The two cited publications indicating that Vitamin D deficiency is associated with more northerly latitudes compare populations within the northern hemisphere against each other (e.g., people living at higher latitudes within the northern hemisphere seem to have higher rates of vitamin D deficiency than those living at lower latitudes within the northern hemisphere). This review does not appear to be comparing populations at different latitudes within the same hemisphere, but rather it groups all populations within the northern hemisphere and posits that a stronger effect should be seen in this population than in analyses that examine populations in both northern and southern hemispheres. Without a comparison between groups at different latitudes or the exclusion of countries below a specific latitude within the northern hemisphere, it does not appear that the effect seen here could be or should be expected to be different from studies that looked at the global population. As such, the rationale for limiting included studies only to those conducted in the northern hemisphere is lacking.

- o Before commencing this work, one of the aims of this review was to investigate possible seasonal relationships between vitamin D and BPPV. Current literature from countries within the Northern hemisphere has demonstrated that serum vitamin D levels vary according to season, which is why this review restricted searches to only countries within the Northern hemisphere.

- o The authors agree that if the seasonal relationship of BPPV and vitamin D were to be investigated thoroughly, that data relating to latitude (plus month of disease presentation) would be required, as seasonal exposure to sunlight (and hence vitamin D availability) varies by latitude.

- o However, as the review process was undertaken it became clear that this would not be possible due to a lack of published data relating to seasonal presentation of BPPV and vitamin D testing, therefore the review focused only on the other aims which were the relationship between BPPV incidence/recurrence and vitamin D levels as

measured using serum blood testing, which has been shown to demonstrate a relationship independent of seasonality or effect of latitude.

o The protocol for the review was pre-registered and published in PROSPERO prior to undertaking the work, at which time it was not possible to know that focussing searches on the Northern hemisphere would not result in sufficient data related to BPPV and vitamin D seasonality.

• Given the flaws in the search strategies, the lack of a manual review of the publications cited by other systematic reviews, and the lack of an analysis of publication/non-reporting bias, it is difficult to determine if the conclusions reached here are valid. Missing studies may have biased the results in one direction or another. The high degree of heterogeneity seen here also casts doubt on the appropriateness of the meta-analyses conducted here, as it seems doubtful that the differences in results seen here are due entirely to change and not to other underlying differences between studies, and as such, the results of the meta-analyses may not be particularly meaningful (<https://training.cochrane.org/handbook/current/chapter10#section-10-10>).

o Thank you for your constructive feedback. Comments relating to search strategy, manual review of publications, exclusion of non-published literature and heterogeneity have been addressed individually above.

o The protocol for the review was pre-registered and published in PROSPERO prior to being undertaken, and this systematic review and meta-analyses followed the preregistered protocol. Statistician input was used to confirm validity and appropriate strategy for meta-analyses.

Reviewer 4:

Dr. Sameer Sharif, McMaster University

Comments to the Author:

Thank you for the opportunity to review this systematic review and meta-analysis on the association between Vitamin D deficiency and BPPV incidence and recurrence. I was asked to do a statistical review of this study. Please see below for my feedback as an opportunity to further improve upon your work.

Thank you for your feedback.

Methods

Search:

- Why were only studies since 2000 included? Why not earlier ones?

o Searches were limited to ensure that included research was reasonably up to date.

This search strategy was pre-registered and published in PROSPERO prior to undertaking the systematic review.

- I see the logic of why only participants in the Northern Hemisphere included in the analysis?

It would be interesting to include both Northern and Southern hemisphere patients and then do a subsequent subgroup analysis

o Many thanks for this suggestion. The purpose behind restricting searches to the Northern hemisphere was to further investigate the relationship between seasonality, BPPV and vitamin D levels. And this was part of our pre-registered protocol in PROSPERO. Unfortunately, seasonal data was not reported in the literature, and this was unable to be investigated as intended. As a result, differences between the northern and southern hemisphere relating to seasonality are not able to be analysed at this time.

- Was a PRISMA form completed? Could you complete one and include it in the Appendix for review?

o A copy of the PRISMA checklist and PRISMA Abstract checklist were included.

However, they were uploaded as a separate file per journal requirements. We have now included these in the Appendix as suggested.

- I believe RoBANS still requires validation? Did you consider using the ROBINS tool? Cochrane recommends the use of the ROBINS tool.

o The use of the RoBANS tool was described in the systematic review protocol which was pre-registered in PROSPERO prior to undertaking the review. As such, continuation with the RoBANS tool seems most appropriate rather than retro-active

application of the ROBINS tool.

o The authors understand that the ROBINS-I tool is utilised to assess risk of bias in studies relating to the efficacy and/or safety of interventions. There are no interventions assessed in our systematic review, only the association between a condition (BPPV) and a potential risk factor (serum vitamin D levels).

Random Effects vs Fixed Effects

- Random effects should be employed here despite what the I² is. Fixed effects does not simply relate to statistical heterogeneity but also as to how the studies were conducted. If the Methods employed by the study were incredibly similar, an argument can be made to use the Fixed-Effects model. This is not the case here.

o Many thanks for your advice relating to the use of the most appropriate model.

Random effects model has been used for the meta-analyses according to your recommendation. And the manuscript plus figures have been updated accordingly.

GRADE: I do not see that this was utilized

- The Methodological standard for reporting meta-analyses is to use GRADE. Please go on the GRADE website and create a summary of evidence table for free. This will allow readers and reviewers to determine the certainty of the evidence.

o The authors understand that the GRADE tool is utilised to display certainty of evidence relating to healthcare interventions. There are no interventions assessed in our systematic review, only the association between a condition (BPPV) and a potential risk factor (serum vitamin D levels)

o The protocol for this systematic review was pre-registered and published in PROSPERO prior to work being undertaken and did not include use of GRADE as no interventions were being assessed.

Thank you once again for allowing me to review your work. I look forward to the opportunity to review it once again after the above suggestions have been accounted for. Specifically, the major things that need revision is the use of the ROBINS tool and

using GRADE.

Reviewer 5:

Dr. Dharma Raj, ICMR

Comments to the Author:

NA

VERSION 2 – REVIEW

REVIEWER	Sharif, Sameer McMaster University
REVIEW RETURNED	08-Jan-2024

GENERAL COMMENTS	Thanks for your addressing my feedback. Fixed Effects: I'm still seeing mention of this model despite the response to reviewer document saying that random effects was used. Could this be corrected? I still think it would be valuable to search the literature prior to the year 2000. I will leave this up to the Managing Editor to decide. Thank you for your thorough work on this topic.
--

VERSION 2 – AUTHOR RESPONSE

Reviewer: 4

Dr. Sameer Sharif, McMaster University

Comments to the Author:

Thanks for your addressing my feedback.

- Many thanks for your feedback which has helped us to improve our manuscript.

Fixed Effects: I'm still seeing mention of this model despite the response to reviewer document saying that random effects was used. Could this be corrected?

- There is only one reference to the fixed-effect model under the "Synthesis methods" subheading in the manuscript which states: "The random-effect model was employed for the meta-analysis if $I^2 > 40\%$, otherwise the fixed-effect model was to be used instead".

- This terminology is used in our methods to explain the criteria applied to our dataset to determine whether either the fixed-effect or random-effect model would be utilised for metaanalysis.

- Whilst only the random-effect model was used for our meta-analyses due to the heterogeneity of the data, this was not known when the criteria determining which method to apply were formulated.

- Apologies for any confusion, this has now been reworded for improved clarity: "The randomeffects model was to be employed for the meta-analysis if $I^2 > 40\%$, otherwise whilst the fixedeffect model was to be used instead if $I^2 < 40\%$."

I still think it would be valuable to search the literature prior to the year 2000. I will leave this up to the Managing Editor to decide.

- We believe the editor is satisfied with our previous justifications for limiting the literature to

that published after the year 2000 including that the Systematic Review protocol including the date limitations of the search strategy were pre-registered and published in PROSPERO prior to undertaking the systematic review.

Thank you for your thorough work on this topic.